# Assessing the Factors Impacting on the Reliability of Wind Turbines via Survival Analysis—A Case Study

**Samet Ozturk [1], Vasilis Fthenakis [1,\*] and Stefan Faulstich [2]** 

[1] Columbia University, 116th St & Broadway, New York, NY 10027, USA; so2429@columbia.edu
[2] Fraunhofer Institute for Energy Economics and Energy System Technology—IEE, Königstor 59, 34119 Kassel, Germany; stefan.faulstich@iee.fraunhofer.de
[\*] Correspondence: vmf5@columbia.edu; Tel.: +1-212-854-8885

**Abstract:** The failure of wind turbines is a multi-faceted problem and its monetary impact is often unpredictable. In this study, we present a novel application of survival analysis on wind turbine reliability, including accounting for previous failures and the history of scheduled maintenance. We investigated the operational, climatic and geographical factors that affect wind turbine failure and modeled the risk rate of wind turbine failure based on data from 109 turbines in Germany operating for a period of 19 years. Our analysis showed that adequately scheduled maintenance can increase the survival of wind turbine systems and electric subsystems up to 2.8 and 3.8 times, respectively, compared to the systems without scheduled maintenance. Geared-drive wind turbines and their electrical systems were observed to have 1.2- and 1.4- times higher survival, respectively, compared to direct-drive turbines and their electrical systems. It was also found that the survival of frequently-failing wind turbine components, such as switches, was worse in geared-drive than in direct-drive wind turbines. We show that survival analysis is a useful tool to guide the reduction of the operating and maintenance costs of wind turbines.

**Keywords:** wind turbines; reliability; survival analysis; failure; down times; availability

## 1. Introduction

Wind energy is being deployed increasingly and its global capacity has doubled over the last six years [1]. Although the availability of wind turbines has reached 98% level in European wind farms [2], improvements of reliability are still needed. The problem is two-fold, one is the high operation and maintenance (O&M) cost, the other is the lost energy production. Wind turbines are monitored during scheduled maintenances and/or by condition monitoring systems to sustain uninterrupted energy production and to avoid high O&M costs. The scheduled maintenance included visual inspection, non-destructive testing methods such as ultrasound and acoustic emissions, and oil level testing and vibration analysis, while condition monitoring systems consist of pressure, heat and vibration sensors [3,4]. Wind farm operators need to develop either new techniques or new decision support tools for their O&M strategies in order to reach the goal of maximizing energy production while minimizing O&M costs.

The failure of wind turbines is due to multiple factors and often also due to unpredicted problems that deserve further analysis [5]. Our aim is to the investigate factors that impact on wind turbine survival and inform the actions that need to be taken to reduce the potential for failures. It is expected that proactive maintenance would decrease failure rates and the associated magnitude of consequences in wind turbines, but this has not previously been quantified. Furthermore, being prepared to respond to failures with readily available spare parts decreases downtime, thus increasing the availability of wind turbines. Hence, spare parts management can be organized in an optimum way and downtime

can be reduced significantly by means of reliability predictions for wind turbines, as applied in other industries [6].

In this study, we investigate the potential factors affecting wind turbine failure and model the hazard rate of wind turbine failure using survival analysis, considering operational, climatic and geographic factors.

## 2. Literature Review

There are several studies determining the potential causes and failure mechanisms of wind turbine failures. Tavner et al. [7] investigated the impact of wind speed on wind turbine reliability and subassemblies of wind turbines in Denmark, and concluded, by using cross-correlation analysis, that the generator, yaw control, mechanical brake and hydraulic systems are more prone to being affected by weather than other subassemblies. Leite et al. [8] used the Markov process to model wind speed characteristics, turbine failure and repair rates and types of turbine, and evaluated the availability factor for Brazil. Hau et al. [9] discussed the main causes and most commonly affected components in wind turbines and mentioned that proximity to the sea increases the possibility of corrosion and eventually failure of a wind turbine. Faulstich et al. [10] investigated different factors, considering their impact on the failure rates of wind turbines using a reliability ranking method. It was shown that wind turbines located close to seawater and in highlands, with high wind speeds, suffer high failure rates [10]. Fischer et al. [11] proposed reliability-centered maintenance for the wind turbine components that were the main drivers of unavailability when components such as the gearbox, generator, electrical system, and hydraulic system failed. They concluded that vibration is the main cause of the mechanical failure of these wind turbine components. Tavner [12] investigated the impact of weather conditions on off-shore wind turbine operations. He concluded that high wind speed, turbulence and wind gusts lower the reliability of the wind turbine blade, pitch and mechanical drive train, whereas temperature and humidity affected the electrical rather than the mechanical components. Wilson and McMillan [13] produced failure probabilities for offshore wind turbines using onshore reliability data and offshore weather data and applying Markov chains and Monte Carlo simulation. It was concluded that temperature and humidity have a lower impact than wind speed on offshore wind turbine failures [13]. Stafell and Green [14] used actual and theoretical load data for 282 wind farms in the UK and examined whether the turbine age had an impact on the failure rates. They concluded that aging increases either the failure rate or downtime or both, since there was a significant power reduction with age. Perez et al. [15] compared the failure rates and downtime values based on different turbine types and aspects and reported that direct-drive turbines have the highest sum of failure rates than geared-drive wind turbines. Reder et al. [16] proposed a framework to analyze supervisory control and data acquisition (SCADA) data using a priori rule mining and k-means clustering techniques and determined the effects of weather conditions on wind turbine failures. They found that winter is the season in which failure frequencies increase, whereas wind speed did not impact the occurrence of failure.

Slimacek and Lindqvist [17] analyzed the reliability of wind turbines using a Poisson process and survival analysis, consider different factors such as the type of turbine, size of the turbine, harshness of the environment, installation date and seasonal effects, applied to the WMEP (Scientific Measurement and Evaluation Program) database. They concluded that turbine reliability has improved over the years and external factors such as lighting, icing and high wind increased the failure rate by 1.7 times. Mazidi et al. [18] proposed a hybrid methodology based on neural networks and a proportional hazard model (PHM) for the maintenance management of wind turbines. They used SCADA data to develop a model using PHM to determine the stress conditions of wind turbines. However, they did not consider external factors due the data constraints. Carlos et al. [19] applied Monte Carlo simulations for maintenance optimization purposes using a generic failure database and wind speed data from a Spanish database. They concluded that the optimum scheduled maintenance interval should be 113 days instead of the general industrial application of 180 days. Andrawus [20] proposed an optimal

scheduled maintenance interval of 30 days for a $26 \times 600$-kW wind farm, whereas Kerres et al. [21] found that corrective maintenance that is carried out after a failure is a better option for a V44–600 kW turbine.

The cited studies investigate the reliability of wind turbines and generally use the average failure rate per turbine as a metric for the reliability of wind turbines. However, in all these studies the survey period ended before the wind turbines came to their end-of-life [7,8,10,11]; this type of data is called censored data. Therefore, survival analysis that accounts for the censorship is a better fit to evaluate the reliability of wind turbines and assess the factors that affect the failure of wind turbines. Also, to the authors' knowledge, to date no study has applied survival analysis to model the hazard rates in order to determine the variable impact of factors such as wind speed, turbine age, distance to seawater and elevational location on wind turbine reliability, although these variables are mentioned in the different studies discussed above. Furthermore, we did not find any published studies quantifying hazard rates through the use of survival analysis based on the scheduled maintenance and history of failure of wind turbines.

Survival analysis has been successfully used to determine the factors impacting on the reliability of mechanical and civil infrastructure systems [6,22]. The International Energy Agency (IEA) recommends applying survival analysis to investigate wind turbine reliability in order to develop optimum maintenance strategies [23]. Our study shows a first-time application of survival analysis, including new potential-risk-related variables such as the number of previous failures and the history of scheduled maintenance. The results can guide maintenance optimization and spare parts management.

## 3. Methodology

We investigated the survival of wind turbines, subsystems and parts of a subsystem through a combination of three methodologies. Firstly, we applied survival analysis to wind turbines from a systems perspective using selected factors (e.g., climatic regions, elevational location, distance to coast, mean annual wind speed, turbine age, turbine type, number of previous failures and scheduled maintenance history); secondly, we investigated the factors affecting only a critical subsystem; and lastly, we applied survival analysis to frequently failing non-repairable components in order to determine the factors that impact the survival of wind turbine components.

### 3.1. Survival Analysis

Survival analysis is a statistical technique that analyzes time-to-event data. In this study, survival analysis was utilized to investigate the time-to-failure of wind turbines as a system, a critical subsystem of wind turbines, and parts of a critical subsystem.

Survival analysis has the advantage over regular regression methods of dealing with censorship when there is no information regarding the exact time that a failure occurred.

The survival function demonstrates the probability of a turbine surviving beyond time t. The basic equations to define the survival analysis are presented in Equations (1)–(5) [22]:

$$\mathrm{F}(T) = \int_0^T f(x)dx \tag{1}$$

$$\mathrm{S}(t) = \int_T^\infty f(x)dx = 1 - \mathrm{F}(T) \tag{2}$$

$$\mathrm{S}(t) = \exp[-\int_0^T h(x)dx] = \exp[-\mathrm{H}(T)] \tag{3}$$

$$\mathrm{h}(T) = \mathrm{f}(T)/\mathrm{S}(T) \tag{4}$$

$$\mathrm{H}(T) = \int_T^\infty h(x)dx = -\ln[\mathrm{S}(T)] \tag{5}$$

where $T$ is the time to failure, $f(x)$ is the probability density function of having a failure at time $x$, and $F(T)$ is the cumulative distribution function showing that a turbine survives until time $T$. Also, $S(t)$ is the survival function that denotes the probability of survival beyond time $T$, $H(T)$ is the hazard rate that represents the probability that a turbine at time $T$ would fail during the next time interval, and $H(T)$ is the cumulative hazard function.

We identified the probability of failure of a wind turbine at a certain time using the Kaplan–Meier estimator [24] and estimated the cumulative hazard using a Nelson–Aalen estimator [25], while comparing the survival of separate groups of wind turbines by applying statistical tests such as a log-rank test [26], which will be explained in the next sections.

### 3.1.1. Non-Parametric Survival Analyses: Kaplan–Meier and Nelson–Aalen Estimators

The Kaplan–Meier estimator is a non-parametric method that does not make assumptions for any distribution. Equation (6) defines the Kaplan–Meier estimator [24]

$$\hat{S}(t) = \prod_{j:t_j \leq t} \frac{n_j - d_j}{n_j} \tag{6}$$

where $d_j$ is the number of individuals that have an event at time $t_j$ where $j = 1, \ldots, k$; $m_j$ is the number of individuals censored in the interval $[t_j, t_{j+1})$; and $n_j = (m_j + d_j) + \ldots + (m_k + d_k)$ is the number of individuals at risk just prior to $t_j$ [24].

On the other hand, the Nelson–Aalen estimator is a non-parametric method to estimate and plot the cumulative hazard function [25]

$$H_{NA}(t) = \sum_{t_i \leq t} \frac{d_i}{n_i} \tag{7}$$

where $d_i$ is the number of individuals that have an event at time $t_i$ and $n_i$ is the total individuals at risk at time $t_i$.

### 3.1.2. Log-Rank Test for the Significance of Survival

The log-rank test is used to test the null hypothesis that there is no statistically significant difference between two groups in the probability of an event. The test statistic is the sum of $(O - E)^2/E$ for each group where $O$ is observed, and $E$ is the expected number of events. The obtained test statistical value is checked in a Chi-distribution table and the corresponding $p$-value represents the probability of the event occurring by chance [23].

### 3.1.3. Semi-Parametric Survival Analysis: Cox Proportional Hazard Model

The Cox proportional hazard model (PHM), also known as the Cox model, includes a parametric baseline hazard function along with a non-parametric hazard ratio. Cox PHM is [22]

$$h(t, z) = h_0(t)e^{zB} \tag{8}$$

where $h_0(t)$ is the baseline function, z is the variable and $B$ is the hazard coefficient for the variable. The hazard ratio between the two groups ($z_1$ and $z_2$) in a factor can be estimated using Equation (9).

$$HR(t, z_1, z_0) = e^{B(z1-z2)} \tag{9}$$

In this study the main interests were to determine the differences in the survival of wind turbines based on selected factors using the Kaplan–Meier and Nelson–Aalen estimators, and to estimate the hazard ratios of the factors that impact wind turbine failures by applying Cox regression. The calculations and plots for the Kaplan–Meier and Nelson–Aalen estimations and Cox regression were obtained using the statistical software SPSS V.25 [27]. Proportional hazard assumptions were checked graphically by



log-minus-log (survival) against survival time graphs. Cox PHM was applied until only significant factors remained, with p-values less than 0.05. The results of the Cox regression are presented in tables consisting following Cox regression parameters:

- Standard error (SE): the SE of the estimate shows the accuracy of the estimation for the observed value.
- Wald statistic: The Wald statistic is the ratio of the regression coefficient *B* to SE. It is used to evaluate the significance of the B coefficients of factors.
- Degrees of freedom: *df* represents the number of sub-factors that are compared against a factor. For example, design type has two sub-factors, direct and geared, thus the df is $2 - 1 = 1$.
- Significance level (sig.): The probability of the coefficient occurring by chance for a specific factor.
- Exp (*B*): The hazard ratio from the Cox regression is given as Exp (*B*).
- Confidence intervals (CI) of 95%: 95% upper and lower levels of coefficients resulting from the regression.

## 4. Case Study Based on WMEP Data

In this study, the application of survival analysis to determine the factors of wind turbine reliability was demonstrated by a case study of wind turbines in Germany. The survival analysis involved the investigation of wind turbine failures recorded in the WMEP database, which covers wind turbines operated in Germany between 1989 and 2008. Fraunhofer Institute for Wind Energy and Energy System Technology (IWES, formerly ISET e.V.) carried out the "Wissenschaftliches Mess- und Evaluierungsprogramm" (WMEP), a continuous monitoring project initiated and funded by the German government (Faulstich, S.; Durstewitz, M.; Hahn, B.; Knorr, K.; Rohrig, K. Windenergy Report Germany 2008: written within the research project Deutscher Windmonitor, 2009). In total, around 63,000 reports on maintenance and repair measures were collected and form one of the most significant collections of reliability data. The events in the WMEP database include scheduled maintenance, scheduled maintenance with replacement or repair, and unscheduled maintenance with a replacement or repair. The WMEP survey collected O&M data from more than 1500 wind turbines, in this study data from 575 of these were ready to be utilized, covering 6188 turbine years of operation and including 19,242 events involving a repair or replacement.

A participant turbine in this study was defined according to the chosen methodological approach; these are shown in Figure 1. According to the systems and subsystems approach, a participant was a wind turbine with a time interval from either the commissioning date or a start date of a failure to either another start of a failure or the end of a survey. In some cases, in the failure data of systems and subsystems, there were unscheduled maintenances that started before the previous failure was resolved. Therefore, for the application of the survival analysis for non-repairable components, a participant was defined as a wind turbine with a time interval between either the commissioning or the replacement of a component and either the start of a failure of a component or the end of a survey. The wind turbine participant types 1, 2, 3 and 4 shown in Figure 1 are considered for the systems and subsystems approach, and participant types 1, 3, 5 and 6 were used for the component approach, in which censored values were included in participant types 3, 4 and 6. It must be noted that unscheduled maintenance and scheduled maintenance with a repair or replacement, which are regarded as failure in the WMEP database, were combined for the survival analysis. Also, it is noted that no distinction was been made between initial and repeated participants in the analysis.

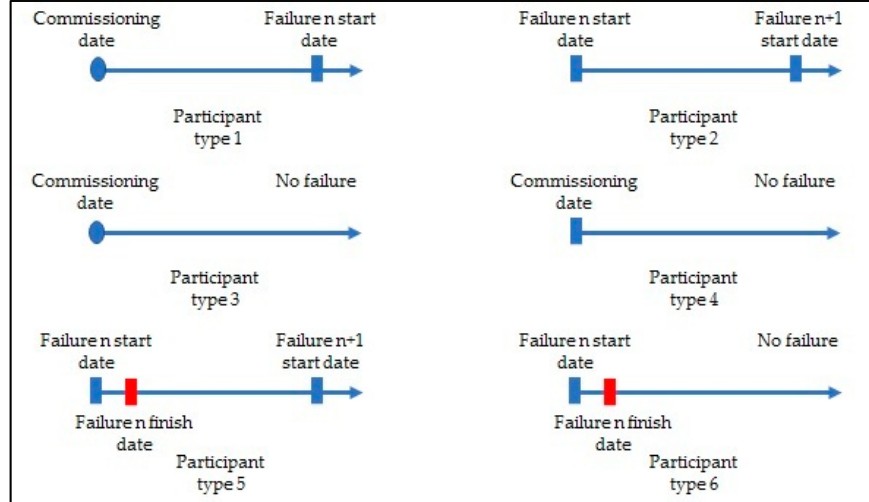

**Figure 1.** Types of participants considered in this study for the survival analysis.

*4.1. Survival Analysis Factors*

Table 1 lists the factors that were considered to have the potential to affect the reliability of wind turbines. The details of the selected factors are provided in the subsections that follow. Some of the factors that may impact the survival of wind turbines—such as production year, turbulence intensity, presence of an online condition monitoring system (CMS), distance to service station, grid quality and energy yield in the analysis—were not taken into account due to data limitations.

**Table 1.** Factors considered in the survival analysis.

| Geographical and Environmental Factors | Operational Factors |
|---|---|
| 1. Koppen–Geiger Climatic Regions: | 5. Turbine age (years): |
| Cfb, Dfb, Dfc | 0–3, 4–14 |
| 2. Elevational location: | 6. Turbine type: |
| High land (>100 m), Low land (≤100 m) | Geared-drive, Direct-drive |
| 3. Distance to coast: | 7. Number of previous failures (NOPF) |
| Coastal (0–20 km), Inland (>20 km) | *Varies* |
| 4. Mean annual wind speed (MAWS)High (>6.25 | 8. Scheduled maintenance history: |
| m/s), Low (≤6.25 m/s) | Yes, No |

4.1.1. Koppen–Geiger Climatic Regions

Koppen–Geiger is a climate classification that has been cited by about 5000 studies in a variety of disciplines [28]. The Koppen–Geiger climatic regions are determined based on annual precipitation and temperature records along with seasonal temperature records; they are based on 12,396 precipitation and 4844 temperature data stations worldwide and employ various temperature and precipitation criteria [28]. In Germany, there are four Koppen–Geiger climatic regions, as can be seen in Figure 2. These are:

- Cfa: Temperate—without dry season—hot summer
- Cfb: Temperate—without dry season—warm summer
- Dfb: Cold—without dry season—warm summer
- Dfc: Cold—without dry season—cold summer

The criteria for the classification of the climatic regions of interest are provided in Table 2 [28]. The first criterion, which is denoted by a capital letter (e.g., C or D), is a climate classification based on the average temperature of the hottest and coldest months. The second criterion is a classification based on the annual precipitation level. The last criterion is based on the summer or winter temperature records.

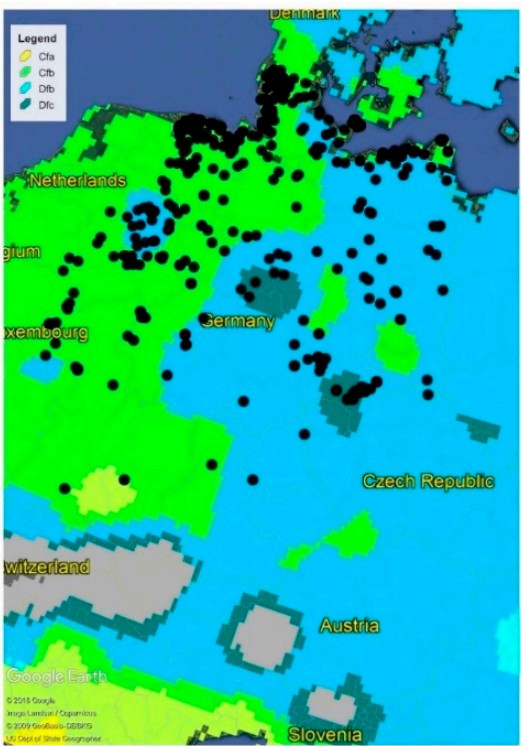

**Figure 2.** Map of the 575 wind turbines (black dots) in different climatic regions in Germany.

**Table 2.** Criteria for the climatic region classification for Germany.

| 1st | 2nd | 3rd | Description | Criteria * |
|-----|-----|-----|-------------|-----------|
| C | | | Temperate | $T_{hot} \geq 10$ & $0 < T_{cold} < 18$ |
| | s | | - Dry Summer | $P_{sdry} < 40$ & $P_{sdry} < P_{wwet}/3$ |
| | w | | - Dry Winter | $P_{wdry} < P_{swet}/10$ |
| | f | | - Without dry season | Not (Cs) or (Cw) |
| | | a | - Hot Summer | $T_{hot} \geq 22$ |
| | | b | - Warm Summer | Not (a) & $T_{mon10} \geq 4$ |
| | | c | - Cold Summer | Not (a or b) & $1 \leq T_{mon10} < 4$ |
| D | | | Cold | $T_{hot} \geq 10$ & $T_{cold} \leq 0$ |
| | s | | - Dry Summer | $P_{sdry} < 40$ & $P_{sdry} < P_{wwet}/3$ |
| | w | | - Dry Winter | $P_{wdry} < P_{swet}/10$ |
| | f | | - Without dry season | Not (Ds) or (Dw) |
| | | a | - Hot Summer | $T_{hot} \geq 22$ |
| | | b | - Warm Summer | Not (a) & $T_{mon10} \geq 4$ |
| | | c | - Cold Summer | Not (a, b or d) |
| | | d | - Very Cold Winter | Not (a or b) & $T_{cold} < -38$ |

* $T_{hot}$ = temperature of the hottest month, $T_{cold}$ = temperature of the coldest month, $T_{mon10}$ = number of months where the temperature is above 10, $P_{dry}$ = precipitation of the driest month, $P_{sdry}$ = precipitation of the driest month in summer, $P_{wdry}$ = precipitation of the driest month in winter, $P_{swet}$ = precipitation of the wettest month in summer, $P_{wwet}$ = precipitation of the wettest month in winter.

Figure 2 shows the wind turbine locations in Germany in the WMEP database that we used in this study. In the WMEP database there are 427 wind turbines and 4526 turbine years in the Cfb region,

122 wind turbines and 1346 turbine years in the Dfb region, 25 wind turbines and 306 turbine years in the Dfc region.

### 4.1.2. Elevational Location

The elevational locations where the wind turbines operate were divided in two categories, namely low land ($\leq$100 m) and high land (>100 m). It must be noted that wind turbines in Germany are not installed at higher altitudes and that the maximum elevation of the turbines considered in this study was 800 m.

### 4.1.3. Distance to Coast

The wind turbines were also divided in two categories based on their proximity to seawater. Turbines with a distance to the coast lesser than or equal to 20 km were described as "coastal", the rest of the turbines were described as "inland".

### 4.1.4. Mean Annual Wind Speed (MAWS)

The mean annual wind speeds at 50 m height for the wind turbine locations was gathered from the Global Wind Atlas for every event [29]. The MAWS values were divided in two categories, namely "low" (lower than 6.25 m/s) and "high" (greater than or equal to 6.25 m/s).

### 4.1.5. Turbine Age

The turbine age categorization was based on the operational years at the initial date of the participant. For example, if the turbine commissioning date was the participant's start date, then the age of that turbine was considered to be 0, if a participant's start date was 370 days after the commissioning date of the turbine, then the age of the turbine was considered to be 1. The age category 0–3 years was considered as "infant" and 4–14 years was considered "mature".

### 4.1.6. Turbine Type

The participants were categorized based on their associated wind turbine design types as "geared" or "direct-drive". Table 3 lists the number of participants associated with each design type.

**Table 3.** Sample data used for survival analysis of the switch component.

| Turbine Model | Time to Failure (days) | Status | Design Type | Climatic Regions | Turbine Age (years) | Distance to Coast | Elevational Location | MAWS |
|---|---|---|---|---|---|---|---|---|
| Model A | 675 | Failed | Geared | Cfb | 0–3 | Coastal | Low | High |
| Model A | 2978 | Censored | Geared | Cfb | 0–3 | Coastal | Low | High |
| Model A | 1572 | Failed | Geared | Cfb | 0–3 | Inland | Low | Low |
| Model B | 3849 | Censored | Direct | Dfb | 0–3 | Coastal | Low | Low |

### 4.1.7. Number of Previous Failures (NOPF)

The NOPF for a participant considered the number of previous failures that had occurred for a turbine or a subsystem, depending on the methodological approach. In the wind turbine full system approach any error was accounted for in the NOPF, while in the subsystem approach only failures in the relevant subsystem were considered in the NOPF. Furthermore, the category intervals varied according to the approach. For the wind turbine system approach, the categorization was divided into five: 0–10, 11–20, 21–30, 31–40, 41+. For the electrical subsystem it was divided into four, namely 0–2, 3–5, 6–8, 9+, since less data points were involved in the subsystem investigation than in the complete systems approach. For the component approach, the categorization was divided into two: 0–1 and 2+.

### 4.1.8. Scheduled Maintenance History

The history of scheduled maintenance categorization in Germany was formed considering the presence of any reported scheduled maintenance without a repair or replacement during the survey period of a participant. Turbines in the WMEP database have routinely scheduled maintenance, as certain measures in wind turbines must follow specific industry standards, such as the IEC 61400-1. However, due to the lack of information in the WMEP database regarding the completion of scheduled maintenance, the scheduled maintenance historical record must be taken as the reporting of scheduled maintenance rather than carrying out them. The IEC 61400-1 standard leads designers through the whole life-cycle of a turbine, from design via operation and maintenance (O&M) to decommissioning. With regard to O&M, the designer has to establish all requirements regarding how to handle special wear parts, safety-related components, greasing, and so on, and also how and when to provide service. A typical period for recurrent inspections has until now been three to six months, while currently designers tend to prolong the period to twelve months, at least for offshore wind turbines. The operator's manual has to state all these requirements and is part of the documents given to a certification body to prove the design assumptions and calculations alongside all accompanying documents. All requirements in the technical documents and certificates will then become part of the mandatory preconditions when the government in charge issues the building and operation permit. Furthermore, the scheduled maintenance history is only considered a factor in system and subsystem approaches. Due to the overwhelming number of participants with a scheduled maintenance history, compared to those participants with no scheduled maintenance history between two component failures, scheduled maintenance was not considered a factor in the component approach of our study.

Table 3 shows sample participants and the associated factors used for the component approach in this study.

### 4.2. Selected Turbine Aspects

Table 4 shows the number of participants based on the methodological approaches discussed above. The most commonly represented direct-drive and geared-drive turbine models, with a 500 kW power production capacity were selected. For the subsystem approach, the electrical subsystem was investigated, as different studies have found it to be the most frequently failing subsystem [2,10,30]. There were 39 geared-drive turbines, adding up to 432 operational years and 70 direct-drive turbines with a total of 733 operational years. Furthermore, fuses and switches, which are the two most-frequently failing electrical subsystem components in the WMEP database, were considered for the component survival investigation in this study.

**Table 4.** Number of wind turbine participants considered in this study.

| Characteristics | Wind Turbine System Study | | Electrical Subsystem Study | | Component Study | | | |
|---|---|---|---|---|---|---|---|---|
| | | | | | Fuses | | Switches | |
| Turbine type | Geared | Direct | Geared | Direct | Geared | Direct | Geared | Direct |
| Number of participants | 1477 | 3334 | 269 | 704 | 47 | 123 | 157 | 126 |

## 5. Results

### 5.1. Wind Turbine System Approach

We used Nelson–Aalen cumulative hazard plots to visualize the differences in factors, since the Kaplan–Meier survival probability graphs do not distinguish differences well when many data points are involved. The most significant distinctions in the cumulative hazard functions in Figure 3 are depicted in the design type and history of scheduled maintenance graphs in Figure 3a,d, respectively, where there was no significant time-dependency in the distinction. On the other hand, the impact of the climatic region, turbine age, distance to coast, elevational location, NOPF and MAWS, seemed

to be time-dependent (as shown in Figure 3b,c,e–h). Thus no definitive conclusions could be drawn related to these factors due to the violation of the Cox regression proportionality assumption. Table 5 summarizes the results of the log-rank tests where one can see that the numerical results support the graphic representation of the design type and the impact of the history of scheduled maintenance on the survival of the wind turbines. From these results, it is inferred that direct-drive wind turbines are significantly more prone to failure than geared-drive wind turbines. A wind turbine that has a history of scheduled maintenance has significantly higher survival than a wind turbine with no scheduled maintenance history, as can be seen in Figure 3d. Moreover, the Cox regression modelling showed that direct-drive turbines were found to be 25% riskier than geared-drive wind turbines, while scheduled maintenance made a wind turbine 2.8 times safer, as can be seen in Table 6. The lower survival rate of the direct-drive wind turbines can be attributed to its lower maturity compared to the geared-drive wind turbines. The higher risk of failure of wind turbines with no scheduled maintenance history was expected, since scheduled maintenance activities are carried out to improve the reliability of wind turbines. On the other hand, the climatic regions shown in Figure 3b seem to not follow any pattern in terms of cumulative hazards, whereas lower turbine ages seem to have a higher hazard rate that diverges in time in Figure 3c. It can be seen in Figure 3e that coastal turbines had a slightly lower hazard rate than inland turbines around the first 500 days of operation and that this was reversed after that time; wind turbines at high elevations had a higher hazard rate than those at lower elevations around the first 700 days, as shown in Figure 3f, this also reversed after that time. It can be inferred from Figure 3g that the smaller the number of previous failures, the greater the likelihood that a turbine will fail again after around 500 days. Also, as observed from Figure 3h, at least for the data considered, MAWS was not a definitive parameter, enabling the conclusion that wind speed affects the survival of a wind turbine as a system.

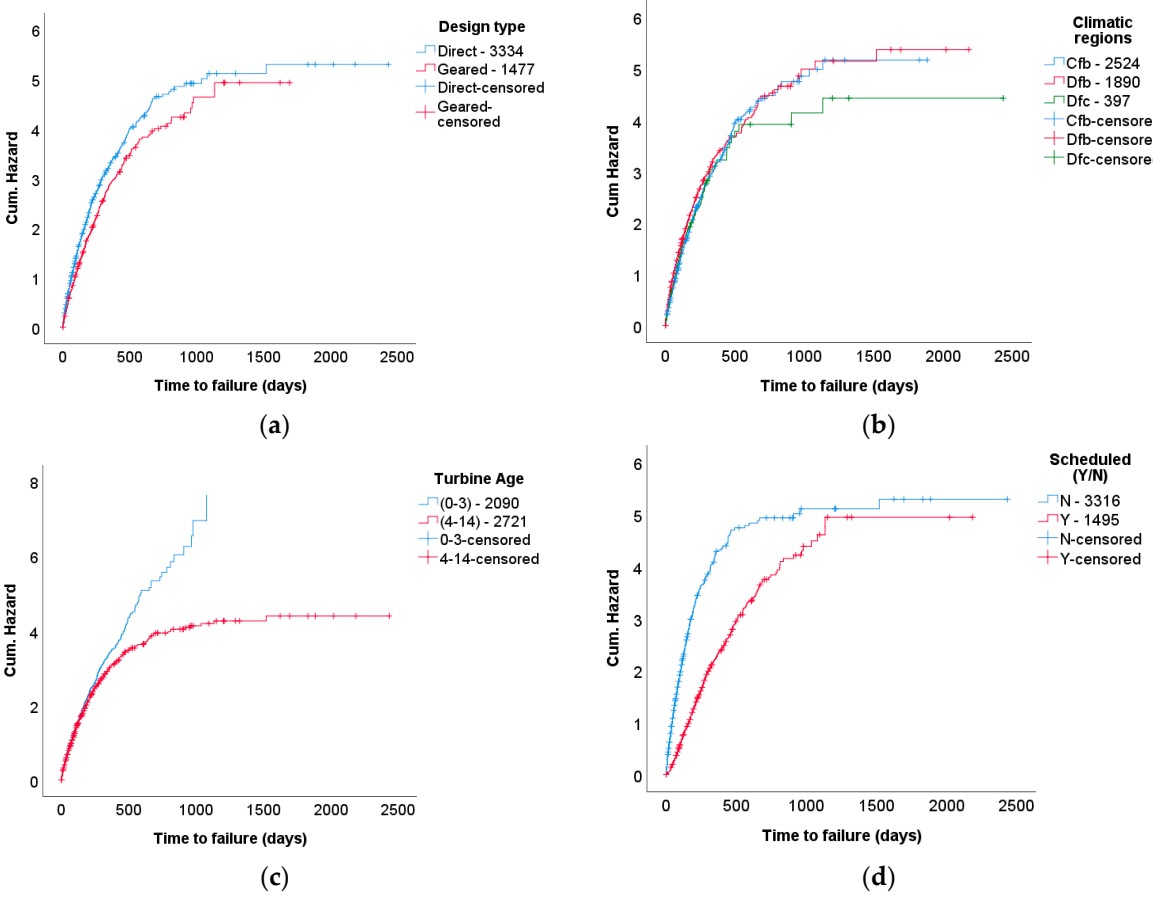

**Figure 3.** *Cont.*

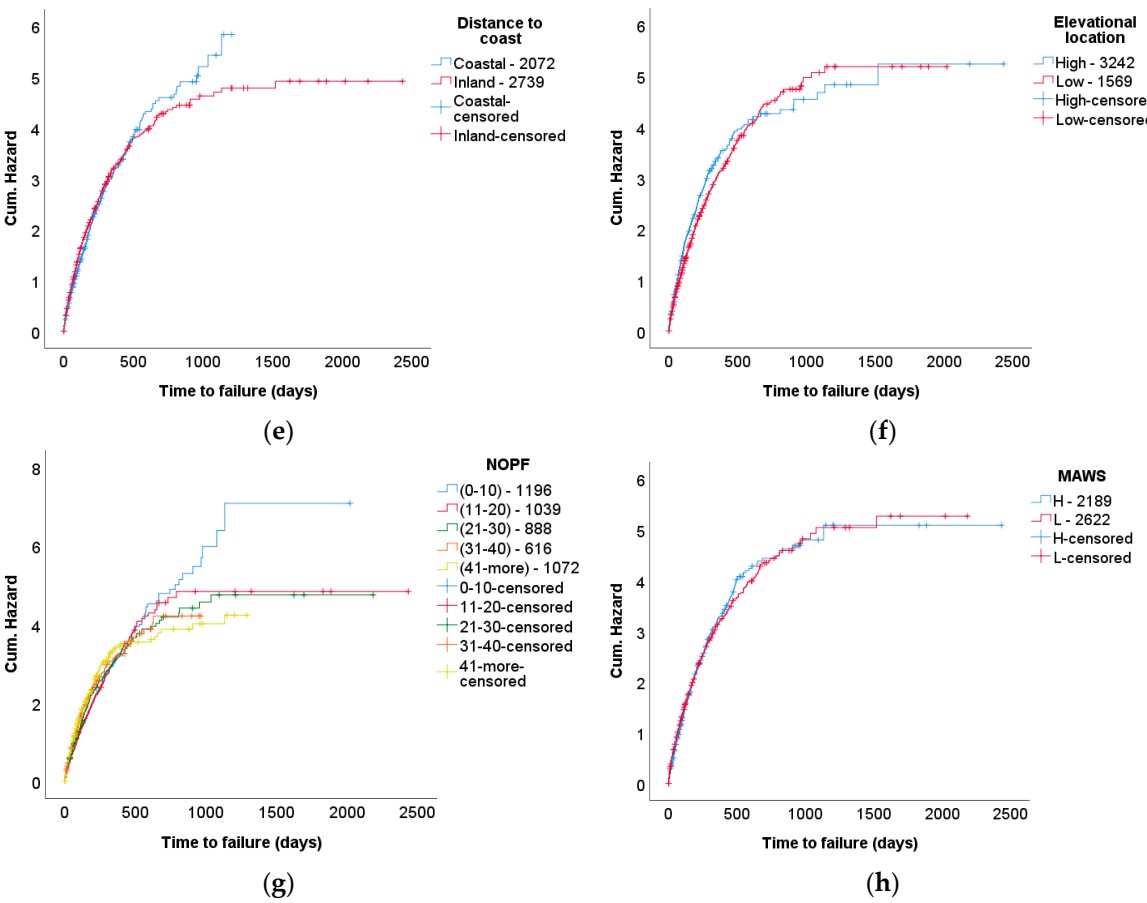

**Figure 3.** Nelson–Aalen cumulative hazard functions of wind turbines based on the operational, climatic and geographical factors. (**a**) Design type (**b**) Climatic regions (**c**) Turbine age (**d**) Scheduled maintenance history (**e**) Distance to coast (**f**) Elevational location (**g**) NOPF (**h**) MAWS.

**Table 5.** Log-rank test results for comparison of the effect of factors impacting on wind turbine system failures.

| Factors | Groups | Test Statistics | |
|---|---|---|---|
| | | Chi-Square | Sig. |
| Design type | Direct vs. Geared | 53.01 | 0.000 |
| Scheduled maintenance history | No vs. Yes | 991.01 | 0.000 |

**Table 6.** Cox regression results for the factors that satisfy the proportionality assumption.

| Factors | B | SE | Wald | Df | Sig. | Exp (B) | 95.0% CI for Exp (B) | |
|---|---|---|---|---|---|---|---|---|
| | | | | | | | Lower | Upper |
| Design type | 0.22 | 0.035 | 39.3 | 1 | 0.000 | 1.25 | 1.16 | 1.34 |
| Scheduled (Y/N) | 1.02 | 0.033 | 935.2 | 1 | 0.000 | 2.77 | 2.59 | 2.95 |

### 5.2. Survival Analysis of the Electrical Subsystem

The survival of electrical subsystems based on the operational, climatic and geographic factors are depicted in Figure 4; notable differences are observed in the history of scheduled maintenance, wind turbine design type, distance to coast and elevational location of the wind turbines. The design type and history of scheduled maintenance have a consistent distinction independent of time, as shown in Figure 4a,d, respectively, while the climatic region, turbine age, distance to coast, elevational location and MAWS are time-dependent factors, as shown in Figure 4b,c,e–h, respectively. Table 7

summarizes the log-rank test results and indicate that there is a significant difference between participants with geared-drive and direct-drive wind turbines, as well as participants with no prior scheduled maintenance and turbines with prior scheduled maintenance. Furthermore, Table 8 provides the Cox regression results, showing that electrical systems in direct-drive wind turbines had a 42% higher risk of failure than those in geared-drive wind turbines, while electrical systems with no scheduled maintenance history had a risk of failure 3.8 times greater than electrical systems with a scheduled maintenance history. It can be inferred from Figure 4c that the impact of turbine age on the survival of electrical systems varies with time. Electrical systems in turbines that were 0–3 years old showed higher survival around their first 1460 days of operation (counting from the participant entry date) than turbines that were 4–14 years old; this subsequently reversed. Although electrical systems in wind turbines in coastal and high elevation locations seem to have lower survival, as can be seen in Figure 4e,f, the Cox regression results presented in Table 8 indicate that these two parameters are not significant. NOPF and MAWS did not satisfy the proportionality criterion for Cox regression, since the survival of electrical systems based on these factors vary with time. Nevertheless, one can see from Figure 4g,h that a smaller number of previous failures and high MAWS increases the survival of electrical systems.

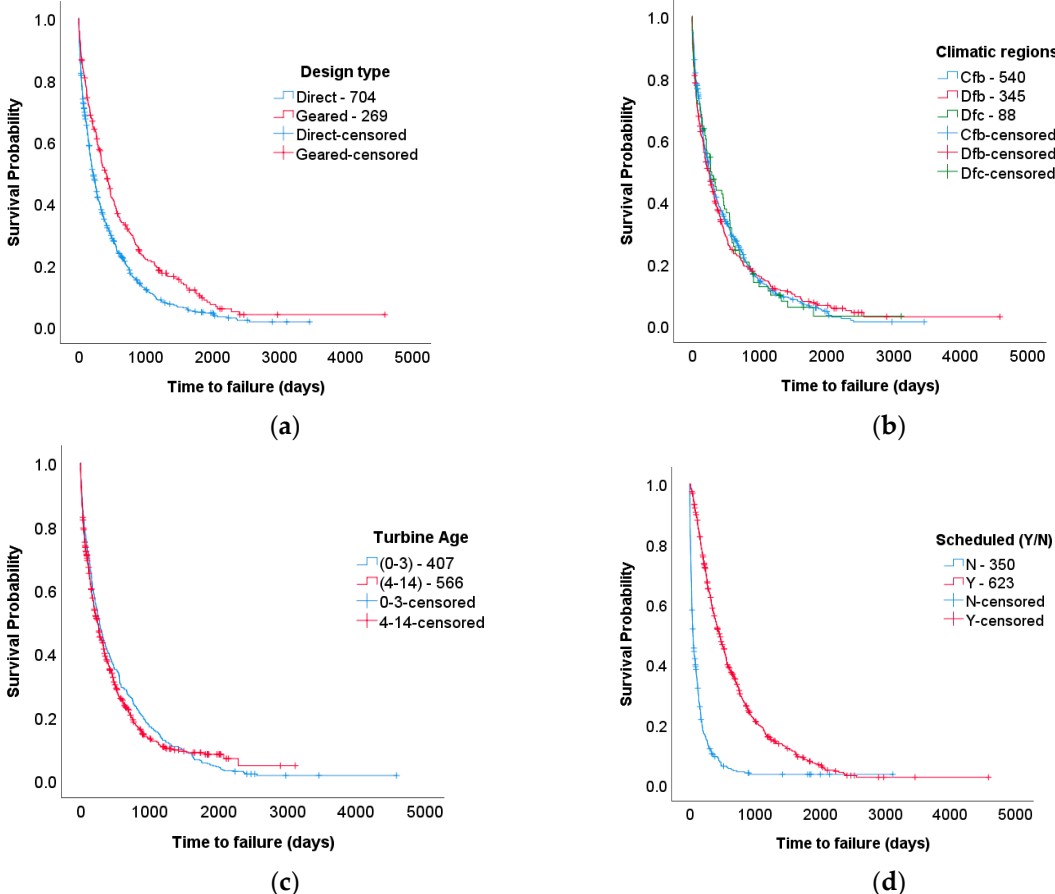

**Figure 4.** *Cont.*

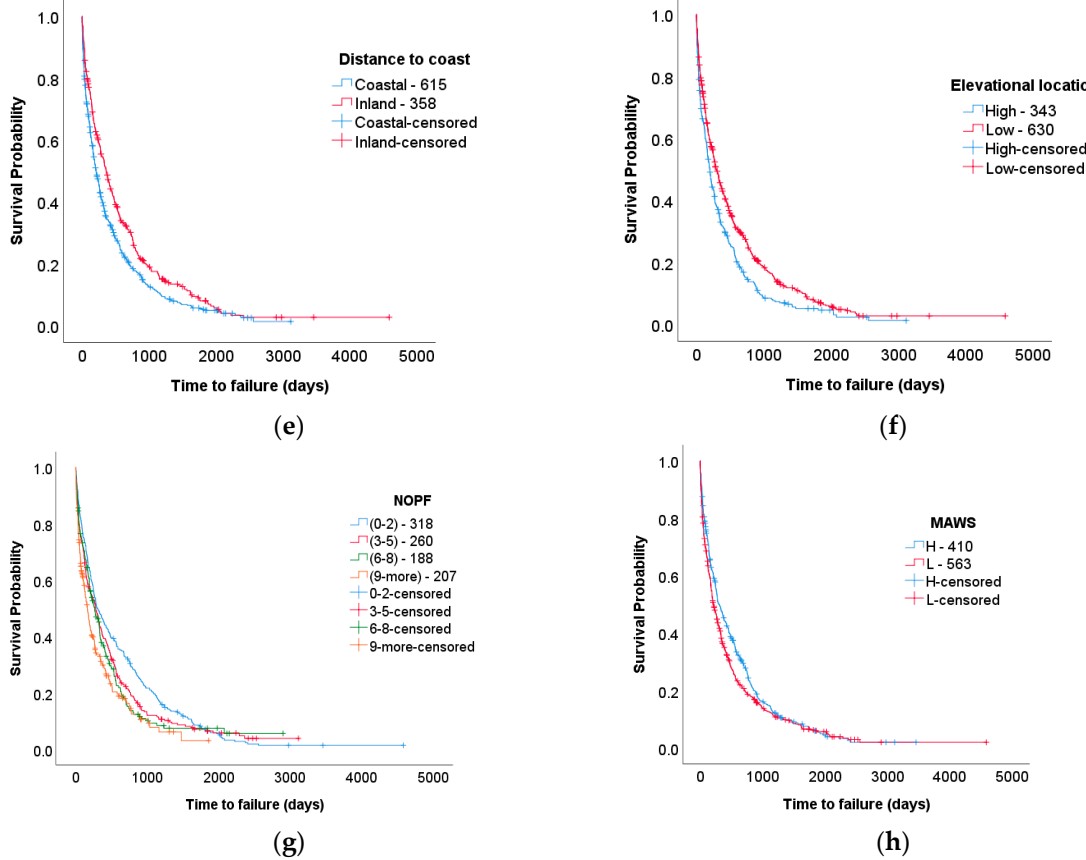

**Figure 4.** Kaplan–Meier survival functions of electrical subsystems based on the operational, climatic and geographic factors. (**a**) Design type (**b**) Climatic regions (**c**) Turbine age (**d**) Scheduled maintenance history (**e**) Distance to coast (**f**) Elevational location (**g**) NOPF (**h**) MAWS.

**Table 7.** Log-rank test results for the comparison of the factors impacting on electrical subsystem failures.

| Factors | Groups | Test Statistics | |
|---|---|---|---|
| | | Chi-Square | Sig. |
| Design type | Direct vs. Geared | 22.77 | 0.000 |
| Scheduled maintenance history | No vs. Yes | 351.76 | 0.000 |

**Table 8.** Cox regression results for electrical system failures.

| Factors | B | SE | Wald | df | Sig. | Exp (B) | 95.0% CI for Exp (B) | |
|---|---|---|---|---|---|---|---|---|
| | | | | | | | Lower | Upper |
| Design type | 0.35 | 0.080 | 18.99 | 1 | 0.000 | 1.42 | 1.21 | 1.66 |
| Scheduled (Y/N) | 1.34 | 0.075 | 319.29 | 1 | 0.000 | 3.81 | 3.30 | 4.42 |
| Elevational location | 0.06 | 0.077 | 0.68 | 1 | **0.408** | 1.07 | 0.916 | 1.24 |
| Distance to coast | 0.04 | 0.087 | 0.20 | 1 | **0.652** | 1.04 | 0.88 | 1.23 |

*5.3. Survival Analysis for Components of the Electrical Subsystems*

Fuses and switches were selected for survival analysis of the electric subsystems of wind turbines, since both components have the highest frequency of failure, as well as being non-repairable components. Fuses provide overcurrent protection, while switches start and stop electrical circuits in a wind turbine [31,32]. They are both discarded after a malfunction since they lose their functionality.

Therefore, it is assumed that NOPF and scheduled maintenance would not have an impact on the survival of fuses and switches, for this reason, these factors were not included in the survival analysis.

### 5.3.1. Survival Analysis for Fuses

Figure 5a,c depict the dependence of survival on the design type and age of wind turbines. The Cox regression results presented in Table 9 show that design type and turbine age are significant factors impacting fuse failure in wind turbines. Fuses in direct-drive wind turbines have a three times higher risk of failure than fuses in geared-drive wind turbines. Since fuse failures occur due to overcurrent, it might be claimed that direct-drive wind turbines might have overcurrent problems more often than geared-drive design turbines. Furthermore, fuses in wind turbines at 4–14 years of operation are 60% more prone to failure than turbines in their first three years of operation, as can be inferred from Table 9. On the other hand, as shown in Figure 5b, the climatic regions did not seem to affect fuse survival, while the distance to the coast, elevational location and MAWS did show some difference, as can be seen in Figure 5d–f, respectively. However, the *p*-values for the representation of significance in the Cox regression showed that the climatic region, distance to the coast, MAWS, elevational location and NOPF were not significant factors, with *p*-values of 0.883, 0.820, 0.802, 0.479 and 0.173, respectively. Thus, these factors were not included in our hazard rate modeling. The proportionality assumption was verified with tests for the design type and turbine age factors.

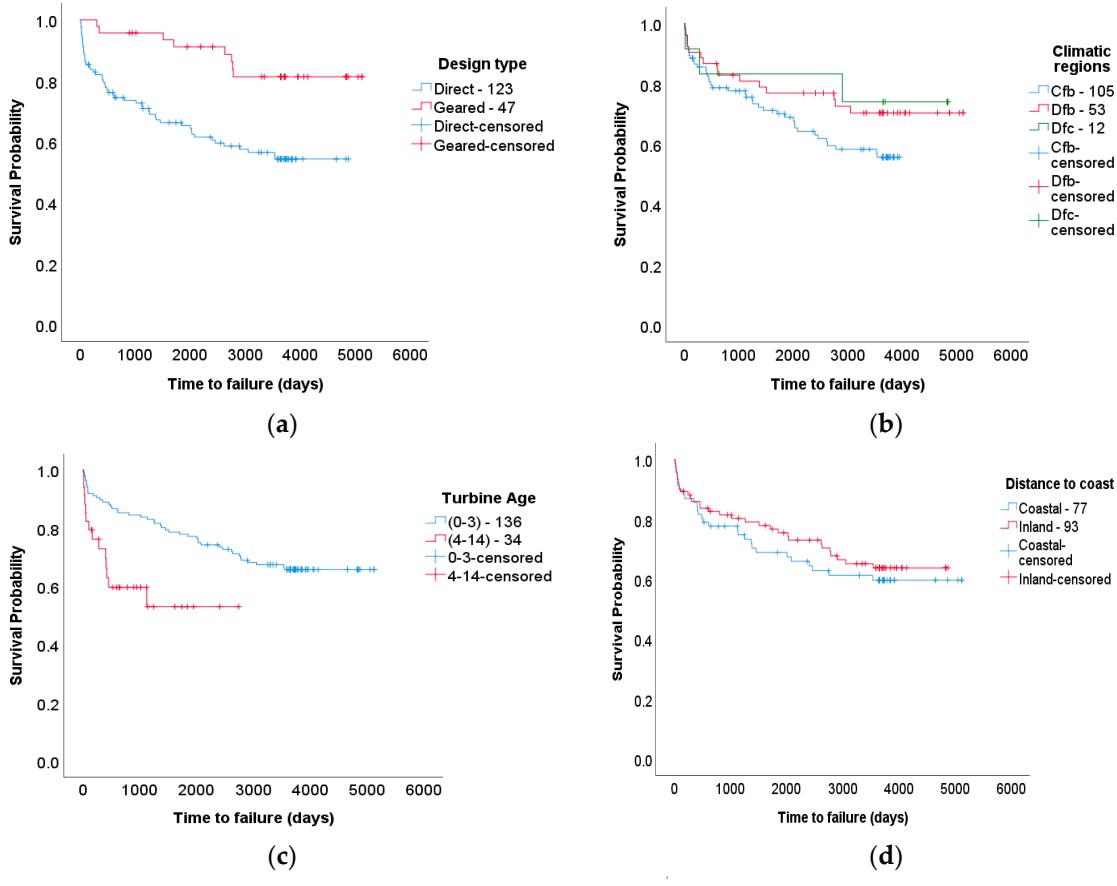

**Figure 5.** *Cont.*

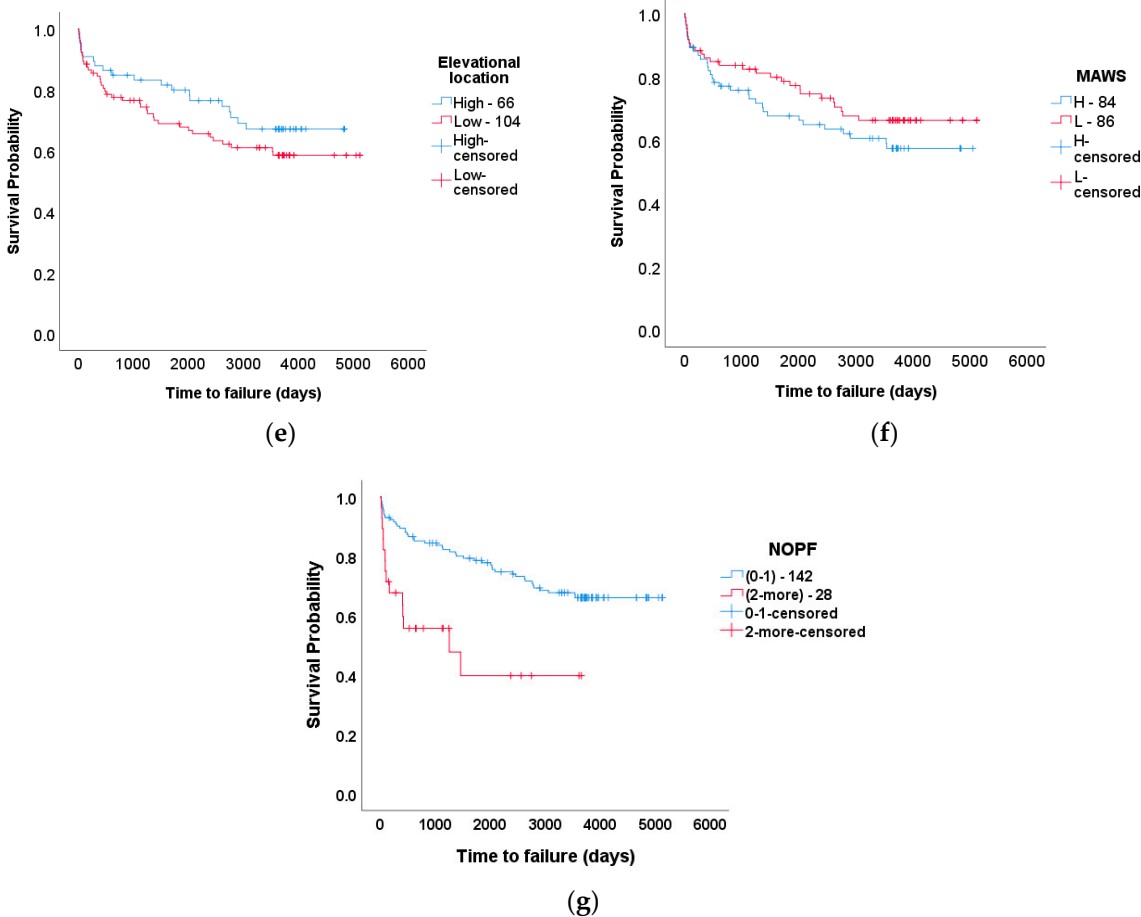

**Figure 5.** Kaplan–Meier survival functions for fuses based on the operational, climatic and geographic factors. (**a**) Design type (**b**) Climatic regions(**c**) Turbine age (**d**) Distance to coast (**e**) Elevational location (**f**) MAWS (**g**) NOPF.

**Table 9.** Cox regression results for fuses.

| Factors | B | SE | Wald | Df | Sig. | Exp (B) | 95.0% CI for Exp (B) | |
| --- | --- | --- | --- | --- | --- | --- | --- | --- |
| | | | | | | | Lower | Upper |
| Design type | 1.13 | 0.381 | 8.73 | 1 | 0.003 | 3.09 | 1.46 | 6.52 |
| Turbine age | −0.95 | 0.329 | 8.35 | 1 | 0.004 | 0.39 | 0.203 | 0.74 |

The hazard rate for the survival of fuses can be written as in the following:

$$HR = \exp[(1.13 \; direct) + (-0.95 \; early \; age)] \tag{10}$$

### 5.3.2. Survival Analysis for Switches

The Kaplan–Meier survival functions of switches in wind turbines are depicted in Figure 6. Obvious distinctions can be observed in design type, distance to the coast, elevational location and MAWS factors, as shown in Figure 6a,d–f, respectively. However, the Cox regression results show that the design type, distance to the coast and MAWS were the factors that have a significant impact on the failure of switches in wind turbines, as can be seen in Table 10. Switches in direct-drive wind turbines have a 66% increased rate of survival compared to the those in geared-drive wind turbines. This may be explained by differences in the material quality or drive design between the two different turbine models. The survival functions of the switches in wind turbines based on their distance to the coast is demonstrated in Figure 6c. The switches in wind turbines that are within 20 km of the coast have a

39% increased rate of survival compared to those located farther from the coast. This can be attributed to more consistent wind speed regimes in coastal regions compared to inland regions. In the coastal regions, switches deal with fewer start–stop cycles, which improves their survival. Similarly, turbines located in regions with high MAWS have a 35% increased rate of survival, as can be seen in Table 10. This can be attributed to the more consistent spinning of wind turbines, which reduces the failure rates of switches. Conversely, turbine age, elevational location, NOPF and climatic regions were determined to be not significant with $p$-values of 0.883, 0.554, 0.520 and 0.088, respectively, and were not included in our hazard rate model. The proportionality assumption was tested, and it was found to be valid with significant factors of $p$-values less than 0.05.

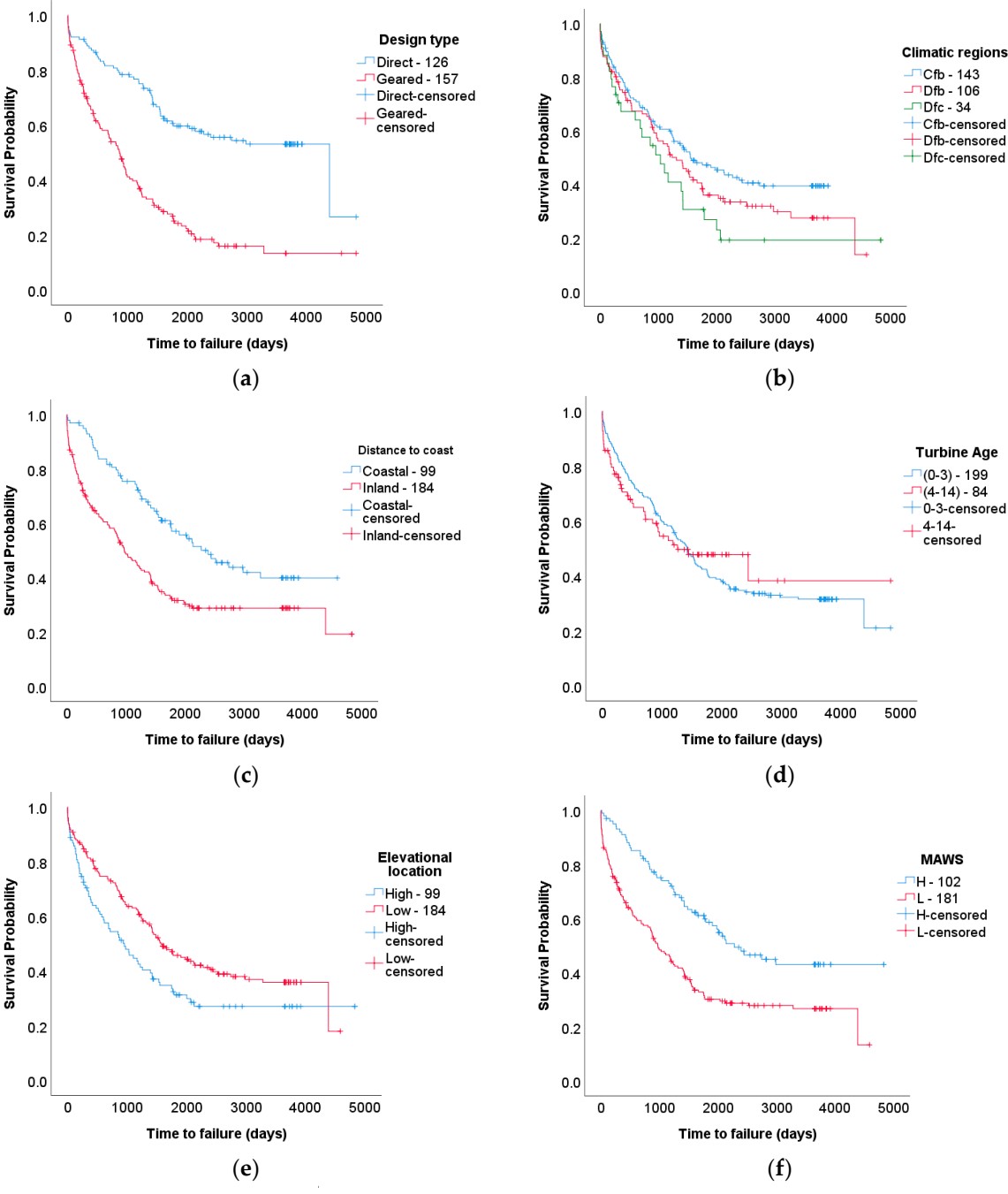

**Figure 6.** *Cont.*

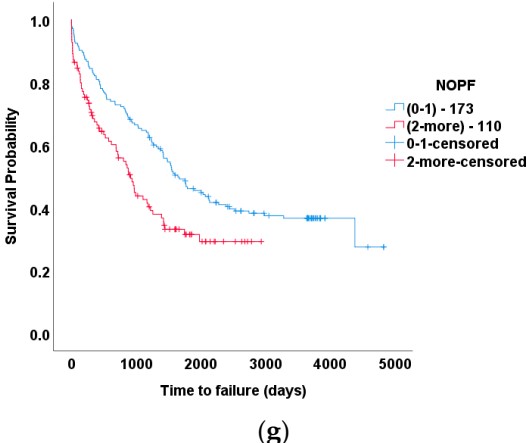

(**g**)

**Figure 6.** Kaplan–Meier survival functions of switches based on the operational, climatic and geographic factors. (**a**) Design type (**b**) Climatic regions (**c**) Distance to coast (**d**) Turbine age (**e**) Elevational location (**f**) MAWS (**g**) NOPF.

**Table 10.** Cox regression results for switches.

| Factors | B | SE | Wald | df | Sig. | Exp (B) | 95.0% CI for Exp (B) | |
|---|---|---|---|---|---|---|---|---|
| | | | | | | | Lower | Upper |
| Design type | −1.08 | 0.167 | 41.48 | 1 | 0.000 | 0.34 | 0.25 | 0.47 |
| Distance to coast | −0.50 | 0.177 | 8.04 | 1 | 0.005 | 0.61 | 0.43 | 0.86 |
| MAWS | −0.43 | 0.178 | 5.89 | 1 | 0.015 | 0.65 | 0.46 | 0.92 |

The hazard rate for the survival of switches can be written as in the following:

$$\text{HR} = \exp[(-1.08 \ direct) + (-0.50 \ coastal) + (-0.43 \ high \ MAWS)] \tag{11}$$

## 6. Conclusions

Survival analysis was carried out considering two novel indicators of the survival of wind turbines, namely the number of previous failures and the history of scheduled maintenance. We identified several risk factors with a definitive impact on the reliability of wind turbines, their electrical subsystems and components of the electrical subsystem using survival analysis. Our results are summarized below:

- Geared-drive wind turbines and their electrical systems were observed to have 1.3- and 1.4- times higher survival rates, respectively, compared to direct-drive wind turbines and their electrical systems. This distinction in survival was also true for fuses, while switches showed the exact opposite trend—switches in direct-drive turbines were less likely to fail compared to switches in geared-drive wind turbines. The geared-drive type of wind turbines might improve the survival of certain components while reducing the survival of other components. For example, the survival of fuses was two times higher in direct-drive turbines than in geared-drive turbines, whereas the survival of switches was reduced by 66%.

- Although the survival probability graphs show some differences between the climatic regions, these were not significant to the survival of wind turbines, the electrical subsystem and components of the electrical systems. However, this significance is related to the number of data points and the relative number of data points among the factors as well as the investigated subsystems and components. The lack of expected significance for the climatic regions in this study may be attributed to data scarcity, specifically for the Dfc region with a cold climate in the summer.

- The impact of turbine age on the survival of turbine systems and electrical subsystems varied with time. However, fuses had a 60% lower survival in the "mature" age group (4–14 years) than in their early years.

- Scheduled maintenance reporting significantly improved the survival of wind turbines; our data and analysis showed a 2.8- and 3.8-times improvement in survival for wind turbines as a system and for the electrical subsystems, respectively. In other words, at any time there was 2.8 times higher probability of survival for a wind turbine and a 3.8 times higher probability of survival for an electrical subsystem with a history of scheduled maintenance than one without such a history.

- Distance to the coast was not found to be a significant reliability factor for wind turbine systems and electrical subsystems. However, the shorter distance to the coast increased the survival of switches by 39%. A potential explanation for this is that wind patterns in coastal regions fluctuate less than ones on land.

- Elevational location was found not to be a significant factor for the survival of turbine systems, electrical subsystems and fuse and switch components. It must be noted that the maximum elevation of the considered turbines in this study was 800 m.

- Although the hazard rate cannot be quantified due to the violation of proportionality, it was found that a high number of previous failures (NOPF) reduced the survival of wind turbines as a system and of the electrical systems compared to low NOPF. It was also found that a high NOPF showed a lower survival rate for wind turbine components, however in order to determine the significance more data are required.

- MAWS was not shown to be a significantly reliable factor for wind turbine systems and electrical subsystems. However, higher MAWS increased the survival of switches by 35%, which can be attributed to more consistent wind patterns.

It must be noted that the WMEP data on which this study was based only covered part of a turbine's life, with the longest period of recording being 14 years from the start of recorded operations. In future studies, the complete life of turbines should be considered in order to obtain more concrete findings. Also, survival analysis can be applied to improve the survival of expensive wind turbine components such as blades, gearboxes and generators, subject to data availability.

**Author Contributions:** S.O., V.F. and S.F. defined the research question and structured the paper. S.O. analyzed the data, performed the literature research and wrote the first draft of the paper. V.F. and SF advised the work and refined the paper.

**Funding:** This research was funded by Ministry of Education of Republic of Turkey. The research work of Fraunhofer IEE for this paper was funded by the German Federal Ministry for Economic Affairs through the WInD-Pool (grant No. 0324031A) project.

**Conflicts of Interest:** The authors declare no conflict of interest.

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
