# Peer review of "Assessing the Factors Impacting on the Reliability of Wind Turbines via Survival Analysis—A Case Study"

_energies, doi:10.3390/en11113034_

Round 1

Reviewer 1 Report

Solid article – congratulations. Just minor suggestions:

Line 178 – WMEP

Please explain the readers that WMEP mean the Scientific Measurement and Evaluation Program, well known in Germany but maybe not so known to a wider audience.

FIG. 1 – The quality is very poor. It looks like a print-screen from a report or other source. If is that the case, please reference the source.

Line 256 – Please highlight to the reader what was the maximum altitude. Wind turbines in Germany are not installed in higher altitude places. I suggest inserting the note present in lines 538-539.

Figures 3 to 6 have a subset of 7 figures staked together that difficults the reading and places the graphics far from the discussion text. For the sake of clarity, the authors are invited to present a different lay-out, for instance, a 4x2 array that will avoid the same figure to be spread in 3 pages as it happens in the case of figures 3 to 6.

line 541 – missing space after (NOPF).

p { margin-bottom: 0.25cm; line-height: 120%; }

Reviewer 2 Report

Overall the paper has good analysis and results.  The following are few of the comments:

1.       Fig.1 need high resolution

2.       Table 1 title must be aligned right

3.       Table 1. Numbering must be corrected

4.       Section 4.2.1 the font on the bullets must align with the text

5.       Must redo the Table 2. Difficult to follow.

6.       Fig.2 need high resolution

7.       Fig.2 title font must align with the text

8.       Table 3 must have units on the corresponding columns such as turbine age (years?). H and L must be expanded

9.       Section 5.1, line 328 and 346 why are they in BOLD?

10.   Table 5 must be aligned right

11.   All equations used must be numbered

12.   Hypen must be removed from the reference numbering

13. The 'fonts' and 'bold' used are inconsistent throughout the paper. Must be corrected.

Reviewer 3 Report

The proposed manuscript is valuable. It is well organized and well written and provides an interesting interpretation of precious data about wind turbine reliability.

I have just a question. Basing on my experience about operation and maintenance of megawatt-scale wind turbines, I have noticed that wind turbines operating since a certain number of years (similarly to the ones that you have considered) can have a turbine condition monitoring system recording mainly gearbox vibrations, but might as well not have it. Do you have this information (having or not having TCM systems) about the wind turbines that you have studied in this work? Do you think it can be an interesting variable for the survival analysis? I have thought this because from your work I have got the following point: if I am a wind farm owner, I would should buy geared-drive wind turbines. Another interesting question, given the considerable cost of turbine condition monitoring systems, in my opinion would be: if I am a wind farm owner, should I equip my wind turbines with turbine condition monitoring systems? Is it an advantageous investment?
